# Prussian Blue: A Nanozyme with Versatile Catalytic Properties

**DOI:** 10.3390/ijms22115993

**Published:** 2021-06-01

**Authors:** Joan Estelrich, M. Antònia Busquets

**Affiliations:** 1Department of Pharmacy and Pharmaceutical Technology and Physical Chemistry, Faculty of Pharmacy and Food Sciences, University of Barcelona, Avda Joan XXIII, 27-31, 08028 Barcelona, Catalonia, Spain; mabusquetsvinas@ub.edu; 2Institute of Nanoscience and Nanotechnology (IN2UB), University of Barcelona, Avda. Diagonal 645, 08028 Barcelona, Catalonia, Spain

**Keywords:** Prussian blue analogues, hybrid nanoparticles, multicatalytic activity, ROS-scavenge, ROS generation, superoxide dismutase-like activity, catalase-like activity, redox potential

## Abstract

Nanozymes, nanomaterials with enzyme-like activities, are becoming powerful competitors and potential substitutes for natural enzymes because of their excellent performance. Nanozymes offer better structural stability over their respective natural enzymes. In consequence, nanozymes exhibit promising applications in different fields such as the biomedical sector (in vivo diagnostics/and therapeutics) and the environmental sector (detection and remediation of inorganic and organic pollutants). Prussian blue nanoparticles and their analogues are metal–organic frameworks (MOF) composed of alternating ferric and ferrous irons coordinated with cyanides. Such nanoparticles benefit from excellent biocompatibility and biosafety. Besides other important properties, such as a highly porous structure, Prussian blue nanoparticles show catalytic activities due to the iron atom that acts as metal sites for the catalysis. The different states of oxidation are responsible for the multicatalytic activities of such nanoparticles, namely peroxidase-like, catalase-like, and superoxide dismutase-like activities. Depending on the catalytic performance, these nanoparticles can generate or scavenge reactive oxygen species (ROS).

## 1. Introduction

Nanozymes are one kind of materials with nanoscale sizes (1–100 nm) and enzymatic catalytic properties. The term “nanozyme” can encompass two different types of materials: (1) enzymes or enzymatic catalytic groups which are immobilized on nanomaterials, and (2) nanomaterials that possess inherent enzymatic catalytic properties [1]. At the present, nanozyme is specifically referred to as nanomaterials with intrinsic catalytic properties [1,2,3]. The first nanomaterial found with enzymatic properties were the iron oxide nanoparticles in 2007 [4]. Since this discovery, the significant advances that nanotechnology, biotechnology, and nanomaterials science have experimented, have allowed emerging of a large number of studies on nanomaterial-based artificial enzymes.

Nanozymes overcome the drawbacks that characterize the natural enzymes, i.e., high cost for preparation and purification, low operational stability, the sensitivity of catalytic activity to environmental conditions, and difficulties in recycling and reusing. Compared with natural enzymes and conventional artificial enzymes, nanozymes have the advantages such as low cost, easy large-scale production, and high stability and durability. However, the most important advantage of nanozymes is their size/composition-dependent activity. It appears that the enzymatic activities of nanozymes are closely related to their size, surface lattice, surface modification, and composition [5]. This allows the design of materials with a broad range of catalytic activity simply by varying shape, structure, and composition. Given these advantages, it is easy to understand the widespread interest that nanozymes have generated and that are becoming powerful competitors and potential substitutes for natural enzymes. In this way, nanozymes have been utilized for practical use in scientific research, biotechnology and food industries, agriculture, environmental treatment, biosensing, disease diagnosis and treatment, antibacterial agents, and cryoprotection against biomolecules in the cell.

According to Huang et al. [1], nanozymes can be divided into two categories: (1) oxidoreductases (oxidase, peroxidase, catalase, superoxide dismutase, and nitrate reductase), and (2) hydrolases (nuclease, esterase, phosphatase, protease, and silicatein).

To date, many nanomaterials have demonstrated to possess remarkable enzyme-like activities. For example, noble and transient metals nanoparticles, carbon nanoparticles (graphene oxide nanosheets, carbon nanotubes, and fullerene), metal oxides, metal sulfides and tellurides, carbon nitride, quantum dots, 2D-nanomaterials with confined single metal and nonmetal atoms, polypyrrole nanoparticles, hemin micelles, and Prussian blue (PB) nanoparticles (PBNPs) [3].

In this review, we will focus on the enzyme-like properties of PBNPs and their applications in this field for biomedicine development. PBNPs have the potential to mimic multiple enzymes in a natural cascade-like system (peroxidase (POD); catalase (CAT); and superoxide dismutase (SOD)). Moreover, the ability to mimic the three indicated antioxidant enzymes makes PBNPs excellent scavengers of reactive oxygen species (ROS), and, contrarily, in various determined conditions, PBNPs can generate ROS by means of a Fenton reaction.

## 2. Prussian Blue Nanoparticles

PB is a kind of metal–organic framework (MOF) composed of alternating ferric and ferrous irons coordinated with cyanides [6]. PB is an aggregated form of ~ 10 nm nanoparticles (PBNPs) [7]. A usual formulation of PB (the insoluble form) is Fe^III^_4_(Fe^II^(CN)_6_)_3_·*x* H_2_O, where the extent of hydration varies from 10 to 16. PB analogues (PBA) are coordination polymers with the general formula *AM*_a_(*M*_b_(CN)_6_)^2/3^·*x* H_2_O, where *A* is an ion (K^+^, Na^+^, NH_4_^+^) alkali ion, and *M*_a_ and *M*_b_ are cations in oxidation state +2 or +3. PB is a commonly used dye (it was the first of the modern pigments, discovered by Diesbach in 1706 [8]) but its applications are not limited to the artistic field but affect many other fields [9]. The development of PBNPs nanoparticles has involved an upturn of their use in a myriad of biomedical applications [10]. Some of them are based on the photothermal effects of these nanoparticles. Recently, interesting antibacterial features of PBNPs coming from this issue have been described [11]. Such nanoparticles benefit from excellent biocompatibility and biosafety since the insoluble form of PB was approved by the Food and Drug Administration (FDA) for the treatment of exposure with cesium and thallium ions. Moreover, it is found in the List of Essential Medicines of the World Health Organization [12]. Apart from the excellent adsorptive properties [13], facilitated by the mesoporous structure, PBNPs have shown catalytic activities, due to the iron atoms that act as metal active sites for catalysis. The different states of oxidation are responsible for the multicatalytic properties of these nanoparticles: one can find a fully oxidized form of Prussian blue (the so-called Prussian yellow, PY), a partially oxidized form (Berlin green, BG), and a reduced form (Prussian white, PW) (Figure 1).

The conversion of one form into other occurs by means of the reactions: PB + 4 e^−^ + 4 K^+^ → PW(1)
PB → BG + 3 e^−^ + 3 K^+^(2)
PB → PY + e^−^ + K^+^(3)

Each oxidation state affords a different color. In the visible spectrum, PB and BG (K_1/3_ Fe(Fe^III^(CN)_6_)_2/3_(Fe^II^(CN)_6_)_1/3_) have two absorption bands at 700 and 480 nm; PW (K_2_Fe^II^Fe^II^(CN)_6_) has no obvious band, and PY (Fe^III^Fe^III^(CN)_6_) presents a band at 420 nm. The multienzyme-like activities of PBNPs (SOD, POD, and CAT) [2] are caused by the abundant redox potentials of their different forms (Figure 2). In this aspect, these nanoparticles are unique since can exhibit simultaneously the indicated three enzyme mimetic activities. Moreover, PBNPs present some advantages that are absent in other nanozymes. In this way, PBNPs can protect ischemia-injured neurons in a rat model due to the antioxidative activity of PBNPs that can eliminate excessive reactive oxygen species (ROS) [15].

There are several ways to obtain PBNPs [16,17,18]. Some of them are easy, relatively cheap, and environmentally friendly. As indicated, the size of such nanoparticles is usually reported to be over 10 nm [7]. Recently, ultrasmall PBNPs ~3.4 nm in diameter have been prepared by using an ethanol/water mixture as the solvent and polyvinyl pyrrolidone (PVP) as the surface capping agent [19]. The stability of the PB or PBA nanoparticles plays a key role in determining the suitability for potential utilization. In most applications, PBNPs are dispersed in a liquid medium such as blood and cellular cytoplasm in living organisms. The colloidal stability is of special importance for any kind of nanoparticle, but in the case of the enzymatic properties of PBNPs, it is crucial, since particle aggregation can lead to significant loss of the catalytic activity. A way to achieve stable dispersions is by modification of the nanoparticle surface. The surface modification of PBNPs has been performed with polyethylene glycol, poly(diallyldimethylammonium chloride), chitosan, native proteins, and poly(vinylpyrrolidone) macromolecules [20]. A simple one-step aqueous solution route for preparing biocompatible PBNPs is coating their surface with a carboxylic acid, namely citric or oxalic acid [21]. The ferric ions are complexed by these anions and form a precursor that reduces the rate of nucleation. Citrate-coated PBNPs did not undergo any change in particle size, size distribution, and dispersibility over a period of more than one year. 

At the present, no inherent inhibitor has been reported for PB/PBA nanozymes. However, in other nanozymes such as iron oxide nanozymes, the peroxidase-like activity decreased in the presence of guanidine chloride (GuHCl) [22]. In this case, GuHCl is supposed to interact directly with the iron atoms of the iron oxide resulting in a change in the electron density of iron. Since PB/PBA also contains iron atoms, the existence of inhibitors cannot be excluded. 

## 3. Peroxidase-Like Activity

Peroxidases are a group of enzymes (E.C. 1.11.1.x) that typically catalyze a reaction of the form ROOR′ + 2e^−^ + 2H^+^ → ROH + R′OH. For instance, in the presence of H_2_O_2_, POD can effectively catalyze the peroxidase substrates into their oxidized substrates. Moreover, POD can decompose reactive oxygen species (ROS), which are involved in multiple key oxidative reactions [23]. PB and PBA show POD-like activity in acidic media because of the large area of ferric and ferrous iron available. These compounds exhibit good H_2_O_2_-induced POD mimicking activity including the oxidation of multiple biomolecules such as 3,5,3’,5’-tetramethylbenzidine (TMB), 2,2’-azino-di(3-ethylbenzthiazoline-6-sulfonic acid) (ABTS), *O*-phenylenediamine (OPD), glucose, dopamine, luminol, guaiacol, and NADH [24]. The generally utilized natural POD substrate is TMB which, when oxidized in the presence of H_2_O_2_, presents two absorption peaks at 369 nm and 652 nm. The intensity of the peaks is proportional to the substrate, and enzyme concentration. Similar to the natural enzymes, the catalytic kinetic behavior of PBNPs followed the Michaelis–Menten equation and depended on the substrate concentration, temperature, and pH. The nanomaterial activity is often compared to the natural enzyme, horseradish peroxidase (HRP). 

The Michaelis–Menten kinetics is one of the best-known models of enzyme kinetics. The model takes the form of an equation that relates reaction rate (*v*) (rate of formation of product) to the concentration of substrate, [S]
(4)$$v=v_{\max}\frac{\left[S\right]}{K_{M}+\left[S\right]}$$
where *v*_max_ represents the maximum rate achieved by the system, happening at saturating substrate concentration. The value of *K*_M_ is numerically equal to the substrate concentration at which the reaction rate is half of *v*_max_. *K*_M_ approximates the affinity of the enzyme for the substrate, and smaller the *K*_M_, the larger the affinity. 

PB exhibits high POD-like activity for the reduction of hydrogen peroxide due to high electroactivity. As said above, the oxidized TMB absorbs at 652 nm. In the presence of PB, such absorption could be masked by the huge band of PB (absorption peak at ~700 nm). In this case, the catalytic activity can be monitored at 450 nm, the absorption band of the fully oxidized form of the mediator TMB^2+^ [24]. The enzymatic-like activity has been observed in PB (and PBA) alone or combined with other substances forming hybrid composites.

### 3.1. Prussian Blue and Prussian Blue Analogues

Although the group led by Gu and Zhang reported for the first time that PB-modified γ-Fe_2_O_3_ nanoparticles had POD-like activity [25], the first study about the use of PB alone as nanozyme was performed by Zhang et al. [26], which observed a catalytic activity towards the hydrogen peroxide-mediated oxidation of classical peroxidase substrate ABTS to produce a colored product. The catalysis followed Michaelis–Menten kinetics and showed a strong affinity for H_2_O_2_. Čunderlová et al. [27] determined that cubic crystals of PB of 15 nm diameter presented a high catalytic POD-like activity. The estimated value of *K*_M_ for TMB was 0.76 mmol L^−1^, and for H_2_O_2_ was 840 mmol L^−1^ (for the same substrates, HRP presented values of 0.147 and 790 mmol L^−1^, respectively). Moreover, in this study, the surface of PBNPs was biotinylated. These modified nanocrystals showed their use in assays based on the competitive affinity of biotin and human serum albumin. Importantly, biotin-PBNPs retained their catalytic activity towards TMB and H_2_O_2_. As a differential trend, the dependence of the rate of reduction of H_2_O_2_ (*v*_H2O2_) on the concentration of TMB did not fit with the model of Michaelis–Menten. An explanation could be the limiting rate of diffusion of TMB through the biotinylated surface. 

Vázquez-González compared the POD-like activity of four inorganic clusters formed by PB and PBA. They observed the generation of chemiluminescence in the presence of luminol and H_2_O_2_, the catalyzed oxidation of dopamine to aminochrome, and the catalyzed oxidation of NDPH to NAD^+^ by H_2_O_2_ [28]. The catalytic activities were different in each compound showing the importance of the chemical structure on the intrinsic properties of nanozymes. PBNPs were most efficient for the oxidation of dopamine and NADH by H_2_O_2_, whereas for the generation of chemiluminescence, the most efficient was the PBA formed by Ma = Fe, Mb = Co (PBCoFe). Komkova et al. synthesized PBNPs at the highest catalytic activity by reducing the mixture of the reagents forming PB using either hydrogen peroxide or a conducting polymer, such as polyaniline [24]. The authors observed that the initial reaction rate of H_2_O_2_ reduction, catalyzed by PBNPs, was linearly dependent on hydrogen peroxide concentration, which had never been described either for POD-like nanozymes or for natural POD. This fact pointed out that H_2_O_2_ activation by the nanoparticles occurred much faster even compared to the natural enzyme. On the other hand, in the absence of H_2_O_2_, no oxidation of TMB was registered. Hence, the employed PBNPs did not display oxidase-like activity (reduction of molecular oxygen). Contrarily, Wang et al. have used PBA nanocages that mimic the oxidase-like activity to the continuous detection of hydrogen sulfide [14]. A PBA described previously (PBCoFe) [28] was employed to catalyze the oxidation of L-tyrosine into dopachrome in the presence of L-ascorbic acid/H_2_O_2_. [29]. In the first step, L-tyrosine is hydroxylated to form L-DOPA, and then, is subsequently oxidized to dopachrome. The mixture L-ascorbic acid/H_2_O_2_ is basic to provoke the hydroxylation of L-tyrosine. The PBA also catalyzed the oxidation of L-phenylalanine to dopachrome. A study conducted by Farka et al. conjugated directly PBNPs with antibodies and applied these nanoparticles in nanozyme-linked immunosorbent assay (NLISA) [30]. The method consisted of the colorimetric readout of the color generated by the oxidized TMB (Figure 3). This technique was used to detect human serum albumin in urine for the diagnosis of albuminuria, as well as for the detection of *Salmonella typhimurirum* in powdered milk, as an example of microbial antigen. When the same antibodies were used in standard sandwich ELISA formats with natural HRP enzyme, similar assay parameters were obtained. This suggested that the assay depended mainly on the affinity of the antibodies for the target analyte and not by the detection step. The easy synthesis from cheap precursors and higher stability of PBNPs in comparison with natural enzymes confirmed the suitability of PBNPs to be used as an enzyme replacement.

Many properties of nanomaterials are dimensionally dependent, which is considered as size effects. The catalytic activity of nanozymes exhibits a size-dependent manner since, for the same volume, nanozymes with a small size expose more active sites. According to this, ultrasmall PBNPs have been demonstrated to possess the highest POD-like activity [19]. The catalytic activity of these nanoparticles was one order of magnitude higher than that of PBNPs obtained by the conventional methods. The values *v*_max_ and *K*_M_ values of such nanoparticles were 2.5 μM s^−1^ and 0.22 mM, respectively, which were much larger than those of PBNPs reported in the literature (i.e., 0.22 μM s^−1^ and 0.34 mM [31]). 

### 3.2. Hybrid Nanoparticles Formed by PB/PBA 

As indicated above, the group led by Gu and Zhang reported for the first time that a hybrid composite formed by PB-modified γ-Fe_2_O_3_ nanoparticles had POD-like activity [25]. It was observed that POD-like activity was increased as the PB proportion increased. This result may reason from that to the more PB providing more ferrous ions as catalysis centers to interact with substrates. The catalytic activity was high, three orders of magnitudes larger than that for magnetite nanoparticles of similar size. With such nanocomposites, an enzyme immunoassay model was established. In this assay, staphylococcal protein A was conjugated on the nanoparticle’s surface and the nanoprobe served to detect IgG immobilized to 96-well plates. Later, this group studied the applications of PB-modified ferritin nanoparticles in biological detection [32]. The same group synthesized PB and gold-modified polystyrene nanocomposites (PS@Au@PB) [33]. In a later study, they remarked that PBNPs showed weak POD activity under neutral conditions [31]. Moreover, in an alkaline environment, they were nearly inactive. More recently, this group inferred that PBNPs may play as ascorbic acid-related nanozyme, which can catalyze the oxidation of ascorbic acid without producing H_2_O_2_ [34]. As the oxidation of ascorbic acid by PBNPs can also be performed in presence of H_2_O_2_, the PBNPs possess ascorbic acid oxidase-like activity besides ascorbic acid peroxidase-like activity.

Cui et al. prepared a nanocomposite by growing PB on the microporous metalorganic framework MIL-101(Fe) [35]. The *K*_M_ values of the composite, PB/MIL101(Fe) with respect to TMB and H_2_O_2_ were 0.127 and 0.0058 mM, lower than those of MIL-101(Fe) for the same substrates (0.490 and 0.620 mM, respectively). The difference was attributed to the fact that the presence of PB in the composite involves more active sites for peroxidase substrates. 

Yang et al. modified PB nanocubes with Co_3_O_4_, a kind of transition metal oxide that exhibits a catalytic activity towards HRP substrates [36]. Such nanoparticles (PB@Co_3_O_4_) exhibited both intrinsic oxidase- and peroxidase-like activities, namely, they could rapidly oxidize TMB with or without H_2_O_2_ at acidic conditions. In this way, a colorimetric method for the detection of glutathione (GSH) by using PB@Co_3_O_4_ nanoparticles as oxidase mimics was developed. GSH produced the fading of the color generated by oxidized TMB (reduction of the absorbance at 652 nm). 

He et al prepared hybrid composites formed by PB and Ti_3_C_2_T_x_, in which this latter acted as a reducing agent, and the multilayer 2D nanostructure could effectively keep the PB nanoparticles away from aggregation. The composite was used as a colorimetric sensor for H_2_O_2_, dopamine, and glucose detection [37]. 

In a study conducted by Sahar et al., a hybrid organic/inorganic hydrogel, which has the potential to mimic multiple enzymes (POD, SOD, and CAT) in a natural cascade-like system was prepared [38]. PB was partially oxidized to Berlin green (BG) and this was mixed with sodium trioxovanadate (NaVO_3_) in the presence of polyvinylpyrrolidone (PVP). After stirring and heating, a hybrid hydrogel (VO_x_BG analogue) was formed. This hydrogel was used as a cascade-like system in the detection of glucose, hydrogel photolithography, oxidation of dopamine, and photocatalytic oxidative degradation of recalcitrant substrates in aqueous media.

An increase in the enzymatic activity was achieved in PBA nanocages doped with molybdenum [14]. This doping successfully tailored the size, morphology, composition, and complex structure of the nanocage. The associate POD-like activity was enhanced by over 37 times compared with pristine PBA.

The POD-like activity of PB/PBA has been used for the detection of several substances such as glucose, glutathione, hydrogen peroxide, or ethanol. In some PBA, which additionally possess oxidase-like activity, this is used to detect glutathione or hydrogen sulfide (Table 1). In the majority of cases, these assays are based on the oxidation of a substrate by hydrogen peroxide. For instance, the aerobic oxidation of glucose in the presence of glucose oxidase (GOx) yields gluconic acid and H_2_O_2_. The oxidation of a substrate by H_2_O_2_ gives a product usually readable by spectrophotometry. In the research by Vázquez-González et al. [28], after the oxidation of glucose, the oxidation of luminol as substrate generates a light of ~428 nm, providing a quantitative readout signal for the concentration of glucose (Figure 4).

Figure 5 illustrates the main reactions implied in the use of PBNPs as POD-like nanozymes. Despite many nanocomposites based on PB/PBA with POD-like activity have been reported until now, their catalytic mechanisms have rarely been investigated. Chen et al. have proved for the first time, by comparing their POD-like activity with that of a series of PBA, in which Fe atoms were replaced by Ni, Cu, and Co [42]. They have demonstrated that the catalytic-like properties can be ascribed to the FeN*_x_* (*x* = 4–6). Inspired by the plane quadrilateral structure of heme, they have proposed that the catalytic mechanism was also similar to that of the heme enzymes (HRP, cytochrome P450). 

From the above, it is evident that PBNPs, as a POD-like nanozyme, can be used as a universal vehicle to build cascades of catalytic reactions. For an efficient action of PBNPs as sensors, such nanoparticles must meet certain requirements such as high surface/volume ratio, compatibility with the other enzymes present in the reaction (in tandem reactions), optimal pH for both enzymes, and the same chemical substrates as natural enzymes. 

## 4. Catalase-Like Activity

Catalases (EC 1.11.1.6) are common enzymes that catalyze the decomposition of hydrogen peroxide into water and oxygen (2 H_2_O_2_ → H_2_O + O_2_). In consequence, the catalytic activity of CAT can be quantitatively analyzed by measuring the concentration of oxygen produced. Due to H_2_O_2_ being a main indicator of inflammation, the evaluation of generated O_2_ is important to evidence any inflammatory process, such as that produced after photothermal therapy. It is habitual for nanozymes to present POD-like activity at low pH and CAT-like activity at high pH. Thus, depending on the pH conditions, H_2_O_2_ can be adsorbed or decomposed on the surface of the nanozymes, and this fact determines their catalytic activity. 

The group led by Zhang and Gu demonstrated that PBNPs could serve as promising CAT candidates in a redox-state-dependent manner under neutral conditions (optimum pH = 7.4) [31]. This group observed that after incubating PBNPs with H_2_O_2_ in an alkaline environment the nanoparticles were inactive, as the POD-like activity is concerned, but bubbles of oxygen appeared in the solution. The amount of gas increased at higher pH levels. The group verified that the CAT-like activity was linearly dependent on the concentration of PBNPs. This study is extremely important because it demonstrated why PB can show POD-like or CAT-like activity as a function of pH. PB, as efficient electron transporter, can present one or another enzymatic activity depending on the redox potentials. For instance, at higher pH, the redox potential of H_2_O_2_/O_2_ is very low, and H_2_O_2_ can be more easily oxidized into O_2_. The same group evidenced the effect of pH on the enzymatic activity of PS@Au@PB nanocomposites [33]. Whereas the POD-like activity was maximal at pH 5.2, the CAT-like activity was remarkable in an alkaline buffer.

The incorporation of a chemical element can modify the pH dependence of the CATlike activity of PB. In this way, PB nanocubes doped with platinum prepared by in situ reduction of PtCl_6_^2−^ on the surface of PB showed prominent CAT-like activity at pH 6.5: the dissolved O_2_ concentration rose from 4.41 mg/L for PB alone to 38.25 mg/L for a PB with a high content in Pt [43].

It is interesting to cite the study carried out by Zhou et al. [44]. They designed a tumor-targeted redox-responsive nanocomposite that may combine tumor starvation therapy and low-temperature photothermal therapy for the treatment of oxygen-deprived tumors. The nanosystem was prepared by loading porous hollow PBNPs with GOx. After this, their surface was coated with hyaluronic acid, therefore allowing the nanocarrier to bind specifically with tumor cells overexpressed in the tumor. Once the nanosystem was introduced into the cell by endocytosis, the GOx was released and oxidized the glucose. As the growth of tumors is highly dependent on glucose supply, this provokes tumor starvation. However, the oxygen concentration in most solid tumors is significantly lower than in normal tissues, which would severely limit the catalytic efficiency of GOx. Consequently, the catalase-like activity of PBNPs can be used to catalyze the intratumoral H_2_O_2_ for rapid reoxygenation, which may help to circumvent the tumor-hypoxia-related tissues.

The oxygen produced by PBNPs’ catalysis has been utilized in some applications for the diagnosis of diseases. Yang et al. developed an oxygen bubble nanogenerator for ultrasound (US) and magnetic resonance (MR) imaging [45]. First, PBNPs decomposed H_2_O_2_ into O_2_ in neutral conditions; then, the nucleated O_2_ molecules could act as gas-bubble US contrast agents. When the oxygen concentration reached supersaturation in tissues, the bubbles could be observed, and an enhanced ultrasound image could also be detected by an acoustic measuring system. Moreover, the paramagnetic oxygen bubbles could shorten the *T*1 relaxation time, acting as an MRI contrast agent. In summary, PBNPs could be an excellent ultrasound (US) and magnetic resonance (MR) dual-modality imaging probe for in vitro and in vivo diagnosing H_2_O_2_ with excellent sensitivity and resolution. 

The CAT-like activity has been used to enhance chemotherapy/photothermal therapy. In this way, a possible synergy between catalytic activity exploited as a photodynamic tool and photothermal effects could be spent. As indicated above, when a tumor grows, there are some portions of the tumor with regions where the oxygen concentration is significantly lower than in healthy tissues. These hypoxic microenvironments in solid tumors are a result of the consumption of available oxygen by rapidly proliferating tumor cells. Moreover, cancer cells produce large amounts of lactate, and this acidic environment promotes tumor cell invasion of adjacent non-cancerous tissue. The indicated hypoxia in solid tumors extremely limits the antitumor of photodynamic therapy. However, the tumor microenvironment also presents high levels of H_2_O_2_, and this can be a source of O_2_. The CAT-like activity of PBNPs can deliver oxygen to the tumor. The generated O_2_ can support the photodynamic therapy of tumors and reduce tumor growth [46].

Peng et al. prepared a PB/manganese dioxide (PBMn) nanoparticle coated by erythrocyte membrane to carry doxorubicin and prolong the circulation time in vivo [47]. The generated oxygen from H_2_O_2_ under the catalysis of PBMn relieved the hypoxia of tumors, and its expansion disrupted the erythrocyte membrane coated on the PBMn surface, which accelerated the release of doxorubicin from the nanoparticles. This made such nanoparticles an H_2_O_2_-responsive activated drug release nanosystem and enhanced the chemotherapy by PBMn. In addition, PBMn, as any PB/PBA, has strong absorption in the infrared region, with high photothermal conversion efficiency, and can be used as an ideal photothermal therapeutic agent [48]. 

Similar to the observed POD-like activity, the CAT-like activity is also size-dependent. Ultrasmall PBNPs tend to be more effective and exhibit higher enzyme-like catalytic activity than PBNPs of higher size [19]. The initial catalytic rate of the ultrasmall nanoparticles was 5.3 mg L^−1^ min^−1^ compared to 0.77 mg L^−1^ min^−1^ of PBNPs [31].

## 5. Superoxide Dismutase-Like Activity

Superoxide dismutase (SOD, EC 1.15.1.1) is an enzyme that catalyzes the dismutation of the superoxide (O_2_^•−^) radical into ordinary molecular oxygen and hydrogen peroxide, two less damaging species. The reaction is: 2 O_2_^•−^ + 2 H^+^ → O_2_ + H_2_O_2_. Superoxide is produced as a by-product of oxygen metabolism and, if not regulated, causes many types of cell damage [49].

As indicated, PBNPs display SOD-like activity [14]. Moreover, the H_2_O_2_ produced by the disproportionation of superoxide can be transformed into H_2_O or O_2_ by PBNPs acting as POD- or CAT-like nanozymes. The standard redox potential values of O_2_/ O_2_^•−^ (0.73 V) and O_2_^•−^/ H_2_O_2_ (1.5 V) indicate that the PBNPs are capable of catalyzing the following half-reactions (see Figure 1): 2HO_2_^•^ + 2H^+^ + PB+ 2e^−^ → 2H_2_O_2_ + BG (5)
2HO_2_^•^ + BG → 2O_2_ + 2H^+^ + PB + 2e^−^(6)
2HO_2_^•^ + 2H^+^ + BG + 2e^−^ → 2H_2_O_2_ + PY(7)
HO_2_^•^ + PY → O_2_ + H^+^ + BG + e^−^(8)

The ability of PBNPs to undergo the dismutation of superoxide radical ions can be assessed by the Fridovich assay [50]. In this assay, the oxidation of xanthine by xanthine oxidase generates superoxide radical ions. Such radical ions reduce nitroblue tetrazolium (NBT) giving diformazan. In this way, the reduction of NBT, yellow-colored, by the generated superoxide radical ions implies the formation of the blue-colored diformazan. In the presence of PB/PBA, the generated radicals are totally or partially scavenged reducing the amount of diformazan and, thus, the intensity of blue color. This reduction can be monitored spectrophotometrically (Figure 6) [20].

Another way to verify the ability of PBNPs for scavenging superoxide anions is by using the xanthine/xanthine oxidase system with 5-tert-butoxycarbonyl 5-methyl-1-pyrroline-N-oxide (BMPO) as a spin trap to form the adduct BMPO/OOH^•^. In the presence of SOD or PBNPs, the signal of the electron spin resonance (ESR) spectrum declined [31,51].

In one study, PB was immobilized on amidine functionalized polystyrene latex (AL) particles [20]. The ability of PB and PB-AL materials in dismutation of superoxide radical ions was determined colorimetrically [49]. The PB did not lose the SOD activity upon immobilization on AL. However, because of the inevitable hindrance of some catalytic sites on the surfaces of PBNPs upon attachment to the AL surface, the maximum inhibition values decrease for PB-AL. In this way, *K*_M_ = 2.19 mM for PB in comparison to 2.92 mM for PB-AL, and *v*_max_ = 6.71 × 10^−6^ M s^−1^ for PB and 4.09 × 10^−6^ M s^−1^ for PB-AL. As an important advantage of these hybrid nanoparticles, the authors indicate their high colloidal stability.

The VO_x_BG analogue aforementioned [38] has multiple oxidation states and this confers it a multi-enzymatic activity. The SOD-like activity of this analogue was determined by the transformation of NBT. 

The platinum-doped PB nanoparticles presented an increased enzyme-like activity in comparison to plain PB nanoparticles [42]. The activity was enhanced with increasing Pt content. Under the same condition, Pt nanoparticles presented a slight SOD-like activity, which contributed to the increased enzyme-like activity of Pt-doped PB nanoparticles.

## 6. ROS-Scavenging or ROS-Generating Activity

The term ROS describes reactive oxygen-derived free radicals such as hydroxyl (OH^•^), superoxide anion (O_2_^•−^), nitric oxide (NO^•^), and peroxyl (RO_2_^•^), as well as nonradical reactive oxygen derivatives such as H_2_O_2_. ROS is a double-edged sword because, although the biological antioxidant system controls the levels of ROS, an excess of them leads to oxidative stress damage and dysfunction related to many human diseases (Alzheimer’s disease, Parkinson’s disease, and diabetes). However, ROS can also play a positive role, since they can induce the activation of apoptosis in cancer cells.

PB and PBA can generate or scavenge ROS depending on their catalytic properties. In this way, PB/PBA with POD- or oxidase-like activity generates abundant ROS; PB/PBA with CAT-like and/or SOD-like activity can remove ROS. In both cases, one can find important biomedical applications, as will be indicated below. 

The formation of ROS by PBNPs follows the Fenton reaction [51]. The conventional Fenton process is associated with the electron transfer from Fe^2+^ to H_2_O_2_ to generate highly active OH^•^, which is capable of effectively attacking the target pollutants and decomposing them into harmless species. The reaction pathway can be illustrated by the following equations:Fe^2+^ + H_2_O_2_ → Fe^3+^ + OH^−^ + OH^•^(9)
Fe^3+^ + H_2_O_2_ → Fe^2+^ + OOH^•^ + H^+^(10)
OH^•^ + substrates → intermediates → CO_2_ + H_2_O(11)

Over the years, several approaches have been carried out to use PB/PBA as Fenton reagents to break down various organic contaminants [52,53,54,55,56]. For instance, Liu et al. employed PB as a photo-Fenton-like reagent and investigated its high catalytic efficiency for rhodamine B (RhB) degradation under visible irradiation in neutral conditions [52]. Li et al. performed PB/TiO_2_ micro composites as a heterogeneous photo-Fenton catalyst to enhance the Fe (II) improvement in destroying organic contaminants [53]. Furthermore, the same group also prepared two kinds of PBA, based on Co and Fe, with different iron valence states and examined their heterogeneous photo-Fenton catalytic mechanism and their use in the oxidation of bisphenol [54,55]. Bu et al. used ultrathin nanosheets of PB to achieve the degradation of methylene blue (MB), one of the most used dyes in various industries [56]. In this study, it was found that, by means of the peroxidase-like activity, 73% of MB had been removed within 1 min in the presence of nanosheets. In the same study, cubic PB only removed 25% of MB. These preliminary studies elucidated that PB/PBA are promising alternatives to conventional transition metals and metal oxides for more efficient catalysis. More recently, the study by Wang et al. focused on the structural control and design of micro/nanocrystals and investigated the relationship between structure and performance of PB [57]. To this end, the authors synthesized a series of PB microcrystals with morphology evolution from microtubes to hexapod stars by adjusting the concentration of chloroplatinic acid in the reaction system (Figure 7).

Results showed that the degradation efficiency of RhB was closely related to the morphology of PB, where the pristine PB-cubs only account for 16.6% of RhB removal within 60 min and PB-hpds enhance that to 97.1% under the same conditions. PB-hpss also presented superior catalytic performance compared to PB-cub and other PB intermediates. The reason was the high specific surface areas and adequate exposure of Fe^III^-NC coordination active sites of PB-hpds and PB-hpss.

The hybrid hydrogel constituting VO_x_ incorporated BG analogue complex was explored as a potential catalyst for photocatalytic degradation of organic pollutants for wastewater treatment purposes [38]. Unlike other photocatalysts, this complex hydrogel followed an unusual catalytic mechanism of OH^•^ radical generation, which in this case function as entrapped holes rather than free radicals in the reaction system. This ability accompanied by the simultaneous generation of OO^•−^ contributed to a superior multicatalytic performance of VO_x_BG.

By modulating ROS production, PB/PBA can be exploited for disease therapeutics. For instance, by catalytically generating abundant ROS selectively in a tumor microenvironment, PB/PBA can be used for antitumor therapeutics, since excessive intracellular levels of ROS may damage lipids, proteins, and DNA. However, the antitumor effect can be reverted by PB itself in some conditions. It is known that ascorbic acid (AA) is able to induce cancer cell death by improving the intracellular ROS level [58]. Although iron is considered to be able to reinforce the anti-cancer effect of AA via Fenton reaction, PBNPs, despite being an iron-based nanomaterial, can inhibit the anti-cancer effect of AA. Zhang et al. have inferred that PBNPs may play as an AA-related nanozyme that can catalyze the oxidation of AA without producing H_2_O_2_ [34]. Moreover, this study showed that PBNPs could catalyze the oxidation of AA in the absence and presence of H_2_O_2_, indicating PBNPs possess ascorbic acid oxidase-like activity besides ascorbic acid peroxidase-like activity. Both enzymatic activities are ROS-scavengers.

PBNPs, after being taken up by cells, can be found in different intracellular microenvironments, where may exert differential enzyme-like activities. PBNPs mainly function as SOD-mimetics, high-activity CAT mimetics, and low-activity POD-mimetics when distributed in the cytosol. In lysosomes, PBNPs unlike other iron-based nanoparticles can inhibit the production of OH^•^ [31]. Without the Fenton reaction, PBNPs could effectively eliminate ROS such as hydrogen peroxide, hydroxyl radicals (OH^•^), and superoxide radicals (OH^•^). With these unique performances, PBNPs have great potential as an anti-inflammatory agent to protect cells from ROS damage. To demonstrate the ROS scavenging ability of PBNPs, Zhang et al. established an in vivo inflammation model [31]. Mice with an inflammation induced by lipoproteins were treated with PBNPs. Results showed that PBNPs could protect the animals from oxidative damage and slow down the inflammatory response. Zhao et al. also showed the ROS-scavenging ability of PBNPs when mice with inflammatory bowel disease were treated with PBNPs administered intravenously [59]. Zhao et al. developed a PBA (based on Mn) to treat inflammatory bowel disease in mice with induced colitis [60]. The prepared PBA accumulated at inflamed sites after oral administration. PBA significantly improved colitis via a primary effect on the toll-like receptor signaling pathway without causing adverse side effects.

Zhang et al. studied the ability of hollow PB to scavenger ROS and reactive nitrogen species, such as nitric oxide (NO^•^) and peroxynitrite (ONOO^−^), in a rat model of ischemia stroke [15]. Apart from attenuating oxidative stress, PB also suppressed apoptosis and counteracted inflammation both in vitro and in vivo, thereby contributing to increasing brain tolerance of ischemic injury with minimal side effects.

The reduction of the size of PBNPs obtained by using a mixture of water and ethanol [17] increased the ability to break down H_2_O_2_ to generate oxygen, scavenging free radicals and protecting cells from oxidation. The scavenging ability was related to the results of the catalytic activity of the nanozyme. This activity increased by reducing the size of the nanoparticles due to the fact that the catalytic activity is size-dependent. The catalytic activity of PBNPs mainly emanates from the electron transfer between Fe ions with different valences [31], and since these catalytic sites are mostly situated at the FeN*_x_* (*x* = 4–6 units), the reduction of the size of PBNPs results in the increase of their specific surface area and allows for the maximization of the exposed FeN*_x_* sites.

## 7. Conclusions and Future Perspectives

PBNPs and their analogues have gathered increasing research interest since their first discovery as nanozymes just over ten years ago [25,26] because they exhibit catalytic properties as peroxidase, catalase, and superoxide dismutase. In this review, we systematically summarized the enzyme-like activities and the catalytic applications of PB and PBA. Compared to natural enzymes, PBNPs overcome their main disadvantages such as the high cost of production and purification. Moreover, PBNPs offer improved tolerance to harsh conditions and higher stability. PBNPs have an added value: their different states of oxidation can be changed by regulating external potentials (PB presents low redox potentials) and this fact is responsible for their multicatalytic properties. Depending on the type of enzymatic-like activity, PB/PBA can be used to remove excess ROS from the body to treat ROS-related diseases and decompose the H_2_O_2_ into oxygen to overcome the hypoxia of solid tumors. However, the same PB/PBA can generate abundant ROS and this property serves for the remediation of environmental pollutants.

Although great progress has been made, the development of PB/PBA as nanozymes with desired properties is still limited by some challenges, and an in-depth understanding of the fundamental principles of these nanoparticles as nanozymes remains limited. It is still necessary to define exactly the influence of size, how the surface modification affect the enzyme-like properties beyond acting as a stabilizer, which is the maximal density and thickness of the coating layer to avoid that the interaction between the nanoparticles and the substrate is masked, which elements are optimal to dope the chemical structure, and so on. Finally, in spite of the safety of PB, the preparation of new analogues, or new hybrid nanocomposites should be accompanied by studies about the potential nanotoxicity of these nanosystems on living beings well as the toxicity in the environment. 

## Figures and Tables

**Figure 1 ijms-22-05993-f001:**
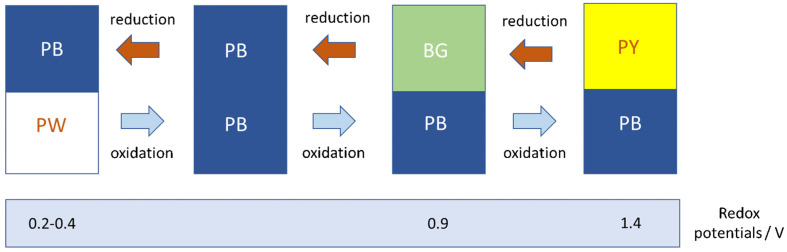
Different forms of Prussian blue depending on the state of oxidation. Each form possesses a characteristic redox potential, which affects the catalytic reaction mechanisms. Reproduced, after modification, with permission from [14].

**Figure 2 ijms-22-05993-f002:**
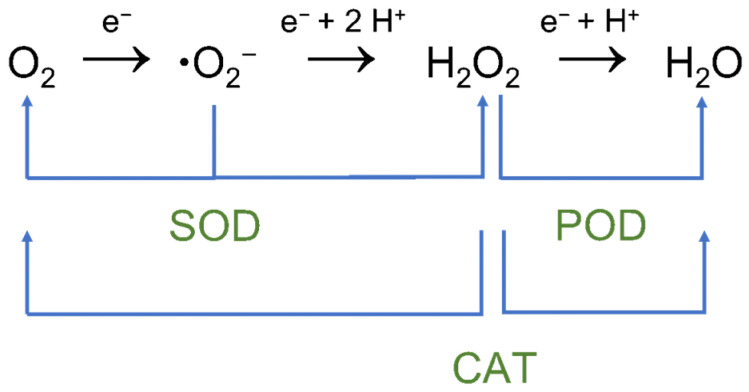
Schematic diagram of the reactions catalyzed by oxidoreductases mimicked by PB and PBA. SOD alternatively catalyzes the dismutation of superoxide into oxygen and hydrogen peroxide; POD breakdown hydrogen peroxide into the water; and CAT breakdown hydrogen peroxide into oxygen and water.

**Figure 3 ijms-22-05993-f003:**
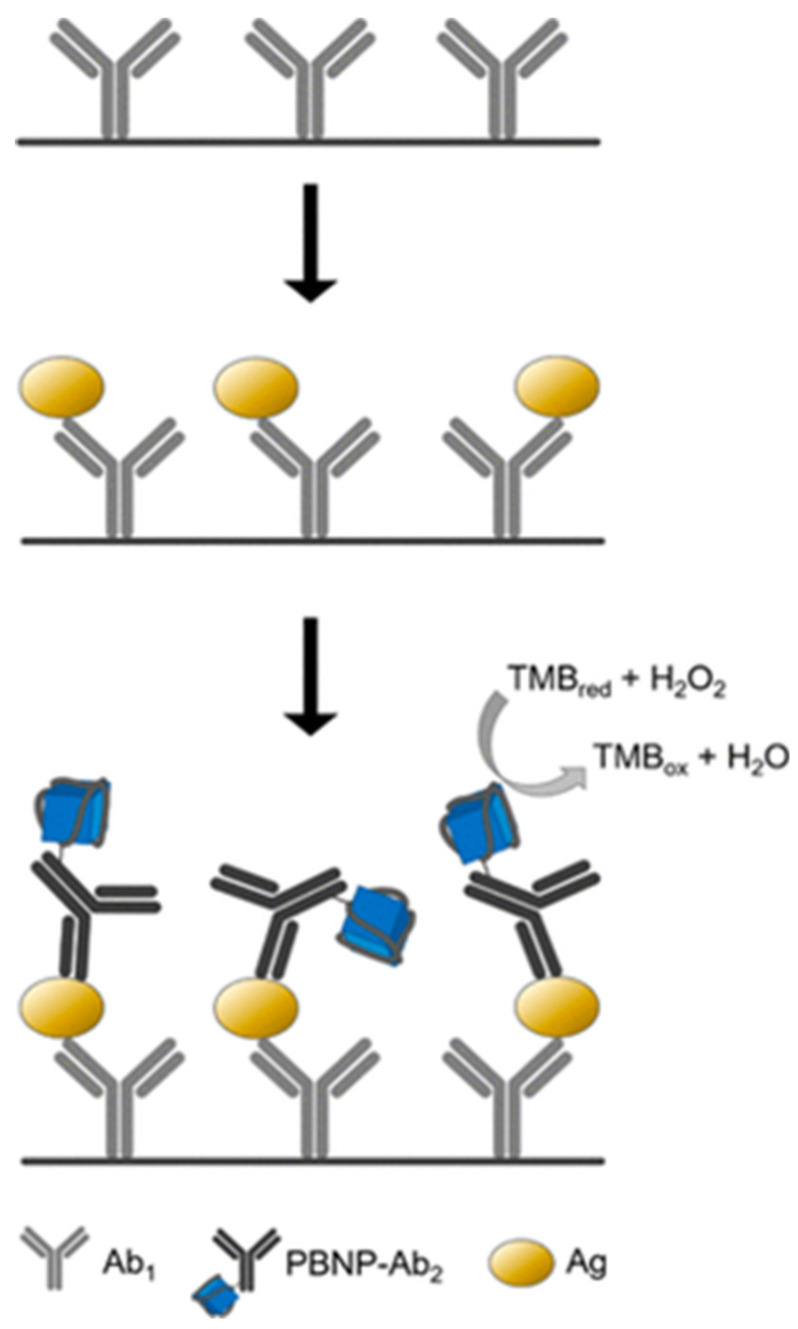
Scheme of PBNPs-based sandwich NLISA. The microtiter plate is coated by capture antibody (Ab_1_), followed by specific binding of antigen (Ag) and formation of sandwich with the detection PBNPs-Ab_2_ label. The colorimetric readout is based on PBNPs-catalyzed oxidation of TMB in the presence of H_2_O_2_. Reproduced with permission from [30].

**Figure 4 ijms-22-05993-f004:**
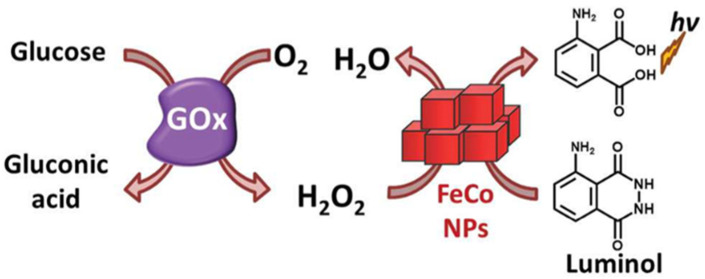
Coupling of the PBFeCo-catalyzed chemiluminescence generation system to the GOx-mediated aerobic oxidation of glucose to gluconic acid and H_2_O_2_ for the development of a glucose sensor. Reproduced with permission from [28].

**Figure 5 ijms-22-05993-f005:**
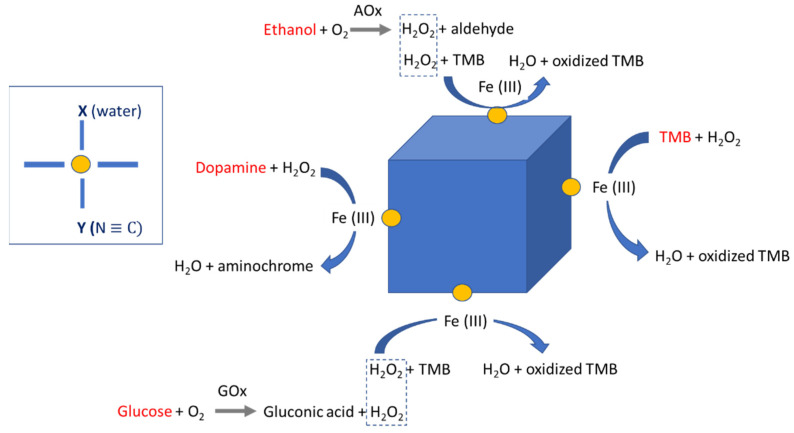
Reactions of the main analytes detected by using PB/PBA as POD-like nanozymes. Inside, the structure of FeN*_x_* band according to [42]; the blue bands represent the macrocyclic assembly of four N ≡ C in the same plane.

**Figure 6 ijms-22-05993-f006:**
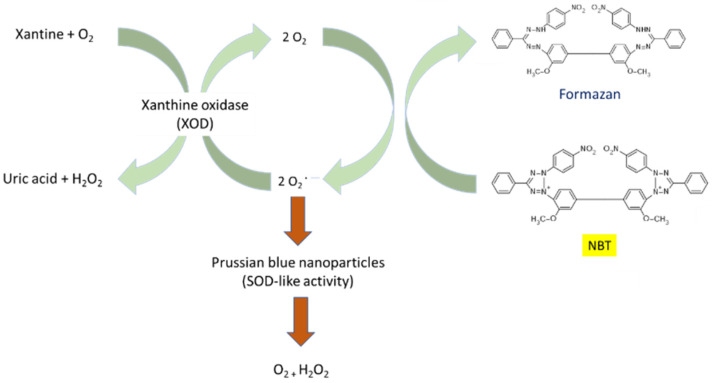
The conversion of xanthine and O_2_ to uric acid and H_2_O_2_ by XOD generates superoxide radicals. The superoxide anions reduce the nitroblue tetrazolium [NBT] to a colored formazan product that absorbs radiation at 560 nm. PBNPs scavenge superoxide anions, thereby reducing the rate of formazan formation.

**Figure 7 ijms-22-05993-f007:**
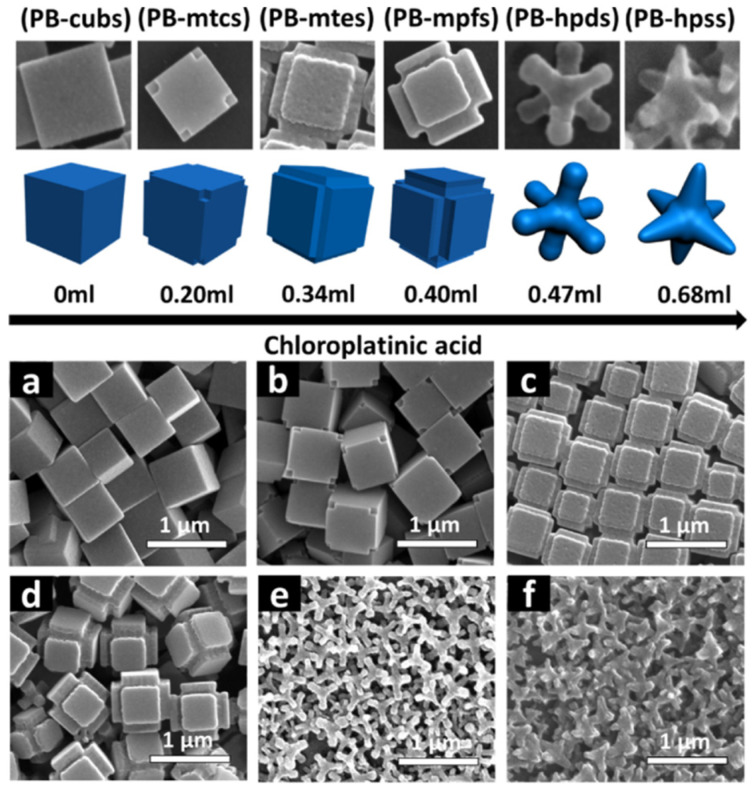
SEM images of PB crystals of different morphology obtained increasing the concentration of chloroplatinic acid. (**a**) PB-cubs: microcubes; (**b**) PB-mcts: PB cubs with truncated corners; (**c**) PB-mtes: PB cubs with truncated edges; (**d**) PB-mpfs: PB cubs with protrusive faces; (**e**) PB-hpds: hexapods; and (**f**) PB-hpss: hexapod stars. Reproduced with permission from [57].

**Table 1 ijms-22-05993-t001:** Prussian blue nanoparticles used as sensors. LOD = limit of detection.

Nanoparticles	Size/nm	Analyte	Linear Range	LOD	Ref
PB	<50	Glucose	0.1–50 μM	0.03 μM	[26]
PB	<50	Hydrogen peroxide	0.05–50 μM	0.031 μM	[26]
Biotinylated-PB	15	Human serum albumin	0.35–???	0.27 μg/mL	[27]
PBA (FeCo)	100–200	Glucose	Not indicated	Not indicated	[28]
PBA nanocages	60	Hydrogen sulfide	0.1–15 μM	33 nM	[14]
Antibody modified PB	24	Human serum albumin	1.2–1000 ng/mL	1.2 ng/mL	[30]
PS@Au@PB	20	Glucose	15.6–250.0 μM	3.9 μM	[33]
Co_3_O_4_-modified PB	~200	Glutathione	0.1–10 μM	0.021 μM	[36]
PB-T13C2Tx		Hydrogen peroxide	2–240 μM	0.4667 μM	[37]
PB		Dopamine	Not indicated	Not indicated	[37]
PB-T13C2Tx		Dopamine	5–120 μM	3.36 μM	[37]
PB-T_13_C_2_T_x_		Glucose	10–350 μM	6.52 μM	[37]
VO_x_BG hydrogel	250	Glucose	1–50 μM	0.046 μM	[38]
Hollow PB	~20	Glucose	0.05–7.3 mM	40 μM	[39]
Hollow PB nanocubes	80	Ethanol	2–500 μg/mL	1.41 μg/mL	[40]
PB/Polypyrrole		Hydrogen peroxide	5–2775 μM	1.6 μM	[41]

## Data Availability

Not applicable.

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
