# Peer review of "Prussian Blue: A Nanozyme with Versatile Catalytic Properties"

_ijms, 2021, doi:10.3390/ijms22115993_

Round 1

Reviewer 1 Report

This paper reviews the enzyme-like properties of PBNPs and their applications for biomedical/biochemical science. As suggested by the authors, the PBNPs and their analogues have gathered increasing research interest since their discovery as nanozymes. The content of this review is timely, interesting, and comprehensive, and its quality is overall high. Therefore, this manuscript will be useful for many readers of IJMS. Thus, I think this paper is publishable in IJMS. Nevertheless, in my opinion, the following minor points should be considered before the publication, mainly for improving the readability of IJMS readers.

(1) Balancing chemical equations.

Several chemical equations in this review ((1)-(3), (5)-(8)) seem to be unbalanced. For example, should the equations (1) and (2) be written as follows?

         PB + A+(K+?) + e- → PW  (1)

         3PB → 3BG + A+ + e-      (2)

Please refine on the balancing of chemical reactions.

(2) Inhibitor?

Many enzymes have their inherent inhibitors. Do the PBNPs have inhibitors? If exist, the Lineweaver–Burk plot (the double reciprocal plot for factors of Michaelis–Menten equation) for ‘inhibited’ PBNPs can be compared to no inhibitor to determine how the inhibitor is competing with the enzyme-like PBNPs. Could you discuss this aspect?

Author Response

TO THE REVIEWER 1

We are thankful to the reviewer for the constructive comments that have helped to improve the manuscript. 

  • We have refined the chemical equations (1-3, and 5-8) by balancing them.

  • As indicated now in the manuscript it is not known the existence of inhibitors of PBNPs. However, we indicate the possibility of this existence. We have added a reference where the action of an inhibitor of the catalytic-like activity of iron oxide nanoparticles is studied. Due to the similitude of the catalytic site of PBNPs with the proposed for IONs, this possibility is open.

Reviewer 2 Report

the paper is timing, as PB nanoparticles use is indeed increasing

the review is well organized, but suffers of some limits that should be amendend before publication

1) language should be carefully reviewd:

phrases like:

"Since nanozymes with a small size would expose more active sites because of the higher surface-to-volume ratio, the catalytic activity of nanozymes seems to exhibit a size-dependent manner. "

or

"Moreover, in an alkaline environment, the catalytic activity was nearly inactive"

should be rewritten as they look like nonsense (how can an activity been inactive?) or poorly understandable.

2) no info is given on phototehrmal activity of PB and PBA, in the introduction or below.

this is a quite important feature of PB NP, and some works in which photothermal effects given by PB and PBA NP are exploited are cited in the review (es refs 37  38 40 41),  and thus some words in the introduction should be spent. For example, a very recent review (Appl. Nano 2021, 2(2), 85-97) appeared on antibacterial features of PB NP coming from this effect, it should be cited.

3) Also, some words (or a proper small section?) on the possibile sinergy between catalytic activity exploited as photodynamic tool and photermal effects could be spent: a recent review pinted out how photothermal efficiency and excellent catalase-like activity were promising for photothermal therapy and O2 self-supplied photodynamic therapy of tumors, and this should be cited (Sensors 2020, 20(23), 6905; https://doi.org/10.3390/s20236905)

.

Author Response

TO THE REVIEWER 2

We are thankful to the reviewer for the constructive comments that have helped to improve the manuscript. 

  • We have modified the indicated paragraphs. The rewritten phases are now more understanding.

  • We have added the indicated reference, as well as the use of PBNPs as bactericidal agents.

  • The information about the synergy between catalytic activity and photothermal/photodynamic effects has been included. Moreover, we have added the indicated paper showing this synergy between catalase-like activity and photodynamic therapy.

Reviewer 3 Report

The review entitled “Prussian blue: A nanozyme with versatile catalytic properties” by Estelrich and Busquets provides an insightful view of the current-state-of-the-art in the use of Prussian blue nanostructures as enzyme surrogates. The topic is interesting and it is worth publishing after a revision is carried out. I enclose below a series of suggestions/corrections that should be ammended prior to final acceptance:

1) Given the review-type manuscript, there are general details about PB NPs that are missing: more specific details about the preparation methods described in the literature (i.e. relevant works from among others Yusuke Yamauchi should be included). I would also recommend the article by S. Scott ACS Catal. 2020, 10, 23, 14315–14317.

2) A specific section describing all the types of PB structures reported in the literature accompanied by illustrations of the 3D structures would be highly desirable and helpful to complete the information of the review.

3) A specific and more detailed section gathering all the issues related with the colloidal stability of these structures and the main ligands used for their stabilization should be also included.

4) Section 3: Should this section be renamed as Peroxidase-like and Oxidase-like activity?

5) Section 3: Addtional illustrations accounting for the main reactions evaluated should be included (i.e. TMB, dopamine, glucose, GSH). In addition, the identification of the active sites in PB NPs should be illustrated and maybe compared with active centers of natural enzymes.

6) Section 3.2.; Page 7: The authors should clearly state the requirements for an efficient action of PB NPs as sensors (they normally require tándem reactions in the presence of other enzymes). This information should be more emphasized and explained.

7) Additional minor corrections/typos that should be checked:

Line 14: spacing in the last Word

Line 17: check shown

Line 18: check spacing in catalase like

Line 56: check spacing

Line 57: check spelling polypyrrole

Line 148: check “by the first time”

Author Response

 TO THE REVIEWER 3

We are thankful to the reviewer for the constructive comments that have helped to improve the manuscript. 

  1. Related to the points 1, 2 and 3, we would like to point out that our manuscript is exclusively aimed to review the properties and characteristics of PB as nanozyme; it does not want to deep into the basic features of PB for being already extensively described in the literature. In that aspect, excellent reviews have been published in the last five years, highlighting the manuscript of Yamouchis’s group (now, is the cite number 15) which deeply and extensively explores details about the preparation of PB/PBA and the resulting structures.

  1. To complete the description of the nanoparticles, we have added three new references [15, 16 and 20] and a paragraph reporting the physical stability of PBNPs coated with citrate ions. 

  1. We understand the reviewer’s point, but we prefer the title Peroxidase-like activity because it is an activity-like intrinsic of these nanoparticles, and, moreover, it is shared by both PB and PBA, whereas the Oxidase-like activity is only found in some PBA. 

  1. We have added a new figure (Figure 5) accounting the main reactions based on the peroxidase-like activity. We have included in the figure (as an inset) the supposed active center of PBNPs.

  1. We have indicated that the use of PBNPs as sensors is accompanied in many occasions by the employment of other enzymes (cascade reactions) and this fact involves several requirements.

  1. The additional corrections have been checked and corrected.